# Contemporary Features and Management of Endocarditis

**DOI:** 10.3390/diagnostics13193086

**Published:** 2023-09-28

**Authors:** Shelby Comeaux, Kiara Jamison, Michele Voeltz

**Affiliations:** 1Department of Graduate Medical Education Internal Medicine, Northside Hospital, Lawrenceville, GA 30046, USA; kiara.jamison@northside.com; 2Department of Cardiology, Northside Cardiovascular Institute, Lawrenceville, GA 30046, USA

**Keywords:** infective endocarditis, native valve endocarditis, prosthetic valve endocarditis, cardiac device related endocarditis, cardiovascular implantable electronic device

## Abstract

Infective endocarditis is a rare but devastating disease. Morbidity and mortality rates have failed to improve despite new technological advances. The disease has evolved over time with new significant populations at risk—most notably those with prosthetic valves or implantable cardiovascular devices. These devices pose new challenges for achieving a timely and accurate diagnosis of infection. While the modified Duke criteria is accepted as the gold standard for diagnosing native valve endocarditis, it has been shown to have significantly inferior sensitivity when it comes to identifying infections related to right-heart endocarditis, prosthetic valves, and indwelling cardiac devices. Additionally, prosthetic valves and cardiovascular implantable electronic devices can exhibit shadowing and artifact, rendering transthoracic echocardiography and transesophageal echocardiography results inconclusive or even normal. Having a keen awareness of the varying clinical presentations, as well as emerging valvular imaging modalities such as F-fluorodeoxyglucose cardiac positron-emission tomography plus computed tomography, promises to improve the evaluation and diagnosis of infective endocarditis. However, indications for appropriate use of these studies and guidance on modern clinical management are still needed.

## 1. Introduction

Infective endocarditis (IE), though a rare disease, continues to carry a significant risk of morbidity and mortality that has remained unchanged over the past four decades, despite advancements in diagnosis and management [1]. This is in stark contrast to improvements observed in other cardiovascular diseases such as myocardial infarction [2]. It is estimated that in-hospital mortality remains at 20% and one-year mortality exceeds 30% [3], a survival rate worse than many cancers [4]. In recent decades, there have been significant changes to the epidemiological and pathophysiological features of infective endocarditis. 

This review encompasses pertinent features of IE, with an emphasis on the changing at-risk patient population groups. The average patient age affected by IE has doubled; a significantly higher proportion of infections involve prosthetic valves and indwelling cardiac devices, and across North America and Europe, Staphylococcus aureus and healthcare-associated disease is now the most common cause of IE [4]. To conclude, we discuss the challenges of diagnosing modern IE, advance imaging modalities, as well as the management of IE in the setting of indwelling cardiac devices.

A literature search using the PubMed database with results from 2013 to 2023 and the terms “infective endocarditis” AND “treatment” OR “diagnosis” OR “imaging” or “management” was conducted. Studies included in this review consist of observational studies, clinical trials, and guideline statements for the diagnosis and management of IE published by the European Society of Cardiology and American Heart Association (AHA). Select studies published prior to 2013 were included if they were considered pertinent and without more contemporary findings. 

## 2. Etiology

Infective endocarditis is an infection of the endocardium, heart valve, or indwelling cardiac device. Its presentation is often varied with a wide array of potential intracardiac and systemic sequelae. The most common causative pathogens are *Staphylococcus*, *Streptococcus*, and *Enterococcus*. These organisms account for approximately 90% of cases worldwide. Less than 10% of cases are attributed to HACEK organisms (*Haemophilus*, *Actinobacillus*, *Cardiobacterium*, *Eikenella*, and *Kingella*), fungi, polymicrobial infections, or other Gram-negative bacteria. Healthcare-associated endocarditis accounts for over 30% of cases, with *S. aureus* being the most commonly isolated organism [4]. This finding has important implications, as S. aureus infections are associated with longer lengths of stay, higher death rates, increased hospitalizations, and elevated costs. 

## 3. Epidemiology

The overall incidence of IE is steadily increasing. Between 2000 and 2011, IE incidence in the United States grew to 15 cases per 100,000 people per year, with global estimates up to 10 cases per 100,000 people per year [5,6,7]. The average age of primary infection has nearly doubled since 1980, from 40 years to greater than 70 years, with the highest incidence seen in men from 75 to 79 years old [8]. In recent decades, the proportion of patients with rheumatic heart disease has substantially decreased, whereas the proportion of patients with IE and prosthetic valves or implanted cardiovascular devices has increased [6,9]. The most frequent predisposing cardiac conditions include degenerative valve diseases, congenital heart disease, and indwelling cardiac devices. Infection seldom affects a previously normal heart and typically requires an underlying presence of structural dysfunction. Non-cardiac risk factors include hemodialysis, immunocompromisation, intravenous drug users, diabetes, and neoplastic disease. The longer survival of patients with heart disease and other risk factors is a significant factor for the increased incidence of IE. 

## 4. Clinical Features

The classic features of IE include sustained bacteremia or fungemia, evidence of active valvulitis, peripheral emboli, and immunological vascular phenomena as described by William Osler. However, in clinical practice, many of these features may be absent and presentation is highly variable [4,10]. Fever and heart murmur are the two hallmark features of infective endocarditis [11]. One international prospective cohort study of nearly 2800 patients found the most common clinical findings to be general markers of inflammation and infection, including fever greater than 38 C (present in 96% of patients with definite IE), an elevated ESR (61%), and elevated CRP (62%). Contrarily, they found that less than half of patients are appreciated to have a new murmur and only one-fifth are found to have accentuation of an old murmur [12]. Textbook findings of splinter hemorrhages, Osler nodes, Janeway lesions, and Roth spots are infrequent, each with a prevalence <10%. Presentation may be acute, with findings of shock, sepsis, and a rapidly progressive course, or subacute with indolent features of fever, malaise, chills, and anorexia. Significant complications of IE include heart failure, stroke, and systemic or pulmonary embolization [11]. 

## 5. Evaluation and Diagnosis 

IE may be diagnosed histologically or clinically. Fulfillment of the modified Duke criteria is readily accepted as the gold standard for clinical diagnosis. The framework of the modified Duke criteria includes major and minor criteria provided by microbiologic, echocardiographic, and physical exam assessment. Combined, these criteria provide a diagnostic probability defining cases as definite, possible, or rejected [13]. Studies have confirmed the high sensitivity and specificity of the modified Duke criteria for identifying definite cases of IE [14,15,16,17,18,19,20,21,22,23,24,25]. However, it should be noted that extending these criteria to infections related to prosthetic valves, indwelling cardiac devices, and endocarditis of the right-heart yields significantly inferior sensitivity [1,26]. Blood cultures and echocardiography form the cornerstone of the major modified Duke criteria. The drawing of three sets of blood cultures, from separate venipuncture sites, is recommended before initiation of empiric antibiotics.

Echocardiography plays a pivotal role in the evaluation and management of patients with suspected IE. Echocardiography can allow for the identification of structural lesions and damage as evidenced by vegetations and perivalvular abscesses. Additionally, echocardiographic assessment informs operative and non-operative management through the evaluation of valvular, ventricular, and global cardiac function, as well as assessing embolic risk and secondary pulmonary hypertension [8,9]. The sensitivity of transthoracic echocardiography (TTE) is reportedly 50%, while transesophageal echocardiography (TEE) is estimated at upwards of 90%. However, much of this data is based on trials from over two decades ago and echocardiography has since seen a significant improvement in image acquisition. 

Advantages of initial imaging with TTE include that it is relatively inexpensive, has better resource availability, is a low-risk examination, confers better hemodynamic and functional assessment, and provides ease in obtaining subsequent assessments. As such, current guidelines support TTE assessment for cases with low probability and moderate probability for IE. In these cases, if TTE fails to identify evidence of IE despite continued clinical suspicion or new cardiac complications, follow-up TEE imaging is warranted [10]. High-volume centers may perform TEE directly for cases with high suspicion for IE; however, in practice, many institutions continue to use the sequential approach of TTE followed by TEE due to the advantages outlined above. The diagnosis of IE via TTE in patients with prosthetic valves and cardiovascular implantable electronic devices (CIEDs) is challenging because shadowing and artifact can yield inconclusive or even normal results in a high proportion of cases. Increased spatial resolution with TEE lessens the observed interference and can confer higher sensitivity for identifying IE [24]. Even so, up to 30% of echocardiographic studies evaluating prosthetic valve endocarditis (PVE) continue to be non-diagnostic [13]. Additionally, for patients with CIEDs, many portions of the device cannot be assessed with echocardiography. As such, additional imaging modalities have been proposed as adjunct studies in cases of suspected PVE or cardiac device-related endocarditis (CDRE) with non-conclusive TEE imaging.

Among complementary forms of imaging for PVE/CDRE, one promising modality appears to be F-fluorodeoxyglucose cardiac positron-emission tomography plus computed tomography (FDG PET/CT). In this whole-body scan, a radiolabeled glucose tracer is utilized to identify metabolically active infectious foci. Studies have observed that FDG PET/CT can identify infectious foci even before the development of morphological or structural damage. As echocardiographic studies rely on evidence of structural lesions, PET has shown particular promise in cases of suspected PVE or CDRE with equivocal or negative echocardiograms. Multiple recent prospective studies have found that incorporating findings of 18F-FDG valve uptake as a major criterion increased the sensitivity of the modified Duke score from 52–70% to 91–97% in both PVE and CDRE [21,25,26,27]. Notably, this observation is largely driven by a reclassification of possible cases to definite cases. Sensitivity for the detection of vegetations is often higher for cases of PVE and CDRE than for native valve endocarditis. Sensitivities for CDRE based on FDG PET/CT alone are reportedly 60–100% [28,29,30]. Moreover, as a whole-body scan, FDG PET/CT can identify extracardiac infectious sources and septic emboli, which is critical for patient management [13]. The implications of these findings are significant for surgical management decisions and considerations of device exchange/removal. Early investigations are also suggesting that the degree of FDG uptake positively correlates to disease severity and may reasonably guide management decisions of antibiotic therapy alone versus complete device extraction, though further studies are certainly warranted [28,29]. The 2020 American College of Cardiology/American Heart Association (ACC/AHA) clinical practice guidelines for the management of patients with VHD include a 2a recommendation for FDG PET/CT for patients with possible NVE or PVE [6]. The 2015 ESC guidelines incorporated findings of FDG PET/CT as major criteria for IE diagnosis in cases of suspected PVE with inconclusive TEE. With the 2023 ESC update, the guideline indications for FDG PET/CT have been expanded to include detection of distant lesions and sources of bacteria, as well as for monitoring drug-therapy response in patients unable to undergo surgical treatment [31].

With respect to IE, multi-slice CT (MSCT) has largely been used preoperatively to evaluate significant coronary artery stenosis. However, growing evidence has demonstrated its use for morphological and functional valvular assessment. Its diagnostic value for IE remains controversial. Concerning the detection of vegetations, MSCT demonstrated significantly lower pooled sensitivity than TEE (86% vs. 94% *p* < 0.001) and no difference in specificity [32]. Moreover, when added in adjunct to TEE, no significant difference in the detection of vegetations was found between TEE and MSCT compared to TEE alone [33]. Similar findings were observed with the detection of leaflet perforation and paravalvular leakage [31]. However, MSCT has shown superiority in the anatomical assessment of abscesses and pseudoaneursyms [33,34,35]. MSCT demonstrates a sensitivity of 87% compared with 69% for TEE (*p* = 0.04) [32]. Current guidelines propose the use of MSCT for IE on an as-needed basis [6,35]. At this time, the data does not support incorporating MSCT into standard protocol and instead its use in difficult cases or in preoperative planning appears to be the most value-added use.

## 6. Native Valve Endocarditis

Native valve endocarditis (NVE) has a reported incidence of 2 to 10 cases per 100,000 people per year [4,36]. For NVE to propagate, vascular endothelial injury or injury to the endocardium must occur. Matrix molecules are then available to adhere with fibrin and platelets to form a ‘sterile’ vegetation. This sterile vegetation then serves as a nidus for circulating bacteria. In the absence of underlying cardiac conditions such as bicuspid aortic valve, degenerative valvular disease, ventricular septal defect, rheumatic disease, or aortic stenosis the immune system typically generates a robust response to the microbial insult [11]. Although rheumatic heart disease is the most common predisposing condition in developing countries, structural heart conditions and intracardiac devices are the more frequent causes in developed countries [11,37]. Other risk factors include IV drug use, diabetes, poor dentition, chronic liver disease, hemodialysis, immunocompromised state, and neoplastic disease [11].

Globally, Gram-positive bacteria are attributed to roughly 80% of NVE cases [11]. *S. aureus* was the responsible organism in 35–40% of cases, streptococcus for 30–40% (20% of those attributed to *S. viridans*; 15% caused by *S. gallolyticus* and other *streptococci*), and 10% of cases with *Enterococci* [4,36,38]. HACEK, polymicrobial infections, fungi, and aerobic Gram-negative bacilli make up the last 5% of cases. Contrastingly, coagulase-negative staphylococci (CoNS) are more commonly attributed to PVE than NVE. A systemic review of global patterns of NVE found that the increased incidence of *S. Aureus* was largely contributed by North America, with no significant change among reports from other continents [1].

Guideline-directed antibiotic therapy is the treatment of choice for NVE and varies slightly by country. Under certain conditions, patients can transition to outpatient parenteral antibiotic therapy (OPAT) following two weeks of inpatient parenteral antibiotic therapy. There are well-accepted surgical indications for patients with NVE: the prevention of systemic embolization, heart failure secondary to valvular dysfunction/perforation, uncontrolled endocardial infection (i.e., persistent bacteremia despite adequate antibiotic therapy or paravalvular extension or cardiac fistulas), and large mobile vegetations [11,39]. In fact, a prospective cohort study by Habib et al. including patients with NVE found, through multivariable analysis, that the failure to perform surgery when indicated was an independent predictor of death [40]. Despite the grave consequences of forgoing surgery when indicated, the literature is not clear regarding the optimal timing, as surgery during the active phase places the patient at increased risk for complications as well. Ultimately, the decision regarding timing has been entrusted to multidisciplinary care teams [38]. Apart from this, prosthetic valve endocarditis and cardiac device-related endocarditis pose a new set of challenges in diagnosis and management in comparison to native valve endocarditis.

## 7. Prosthetic Valve Endocarditis

There has been a significant increase in the incidence of prosthetic valve endocarditis [16]. PVE previously accounted for 5% of infective endocarditis cases, but contemporary studies demonstrate that 20–34% of definite IE cases are PVE [2,17,18]. Mortality rates for PVE patients are also higher, with estimates ranging from 22% to 40% [2]. In comparison with NVE, patients with PVE are older, less likely to use injection drugs, and more likely to have healthcare-associated infection. *Staphylococcus* accounts for approximately 40% of PVE cases, with *S. aureus* now the most common pathogen. Despite similar rates of complications and surgical interventions, in-hospital mortality is significantly higher for PVE than NVE cases [2]. This has largely been attributed to PVE affecting a disproportionately older population with a higher burden of comorbidities. 

PVE is delineated by the timing of presentation. Early-onset PVE is defined as occurring less than one year after surgery; late-onset greater than one year. An Italian retrospective study of approximately 200 persons with PVE found that 43% presented with early-onset vs. 57% with late-onset [19]. Mechanical valves are most likely to have PVE within the first 3 months perioperatively, whereas bioprosthetic valves are more likely to present as late-onset PVE. *Staphylococcus* is the most likely pathogen of PVE regardless of the timing of onset. However, in late-onset, the proportion of staph cases decreases with a rising incidence of *Enterococcus* and *S. viridans*. Nonetheless, early PVE is associated with a high mortality rate, and conservative management (antibiotics) is unlikely to result in a cure. Consequently, the 2023 ESC guidelines recommend complete debridement and new valve replacement within 6 months at a Class I, LOE C [31].

The Partner Trial showed no difference in the incidence of PVE following surgical aortic valve repair versus transcatheter aortic valve repair [20]. However, there is conflicting evidence on whether incidence rates vary significantly by valve type (bioprosthetic vs. mechanical). A Danish 20-year retrospective study found that bioprosthetic aortic and mitral valves were associated with a higher incidence of PVE than mechanical valves, respectively. One US study similarly found an increased incidence of IE among bioprosthetic valves; however, those who underwent bioprosthetic valve replacement were significantly older and more likely to have cardiovascular comorbidities. Another prospective, multicenter study utilizing the International Registry of Infective Endocarditis found that nearly 70% of patients had a bioprosthetic aortic valve and 50% had the presence of a bioprosthetic mitral valve [2]. These findings should be interpreted cautiously as their study designs do not allow for an accurate assessment of causality and have many confounding variables. With the rising incidence of PVE, studies assessing primary PVE risk and IE recurrence by valve type are certainly needed. 

Prior to 2007, antibiotic prophylaxis was routine for patients with prosthetic valves undergoing invasive procedures. A single oral dose of 2 g amoxicillin was recommended for those at moderate or high lifetime risk of acquiring IE. Prophylaxis was directed against *S. viridans* and *Enterococcus* due to their commensal nature of the oropharynx and gastrointestinal/genitourinary tracts respectively. In a departure from previous AHA guidelines, the 2007 update no longer recommended routine IE prophylaxis, but purported it was reasonable for patients with prosthetic valves undergoing specific dental procedures and respiratory procedures at a Class IIa, LOE C. The working group cited the fact that no published data convincingly demonstrated that the administration of prophylactic antibiotics prevents IE. These recommendations appear to be appropriate when held in consideration against the modern profile of PVE, in which fewer cases are reportedly due to *S. viridans* and *Enterococcus*. The ESC similarly recommended anti-streptococcal prophylaxis for patients with prosthetic valves undergoing invasive oral procedures and has since strengthened this recommendation at a Class I, LOE B within the 2023 update [31]. The task force cited the rising European incidence of IE and recently published studies, further substantiating the benefit of prophylaxis in high-risk individuals. The 2023 ESC guidelines also strengthened the recommendation for systemic antibiotic prophylaxis for high-risk individuals, including those with prosthetic valves, to level II recommendation. The primary prophylactic regimen continues to be amoxicillin/clavulanic acid. However, as the incidence of PVE continues to rise, with an increasing proportion from healthcare-associated staphylococcal infections, it raises the question of whether there should be other measures/recommendations directed toward anti-staphylococcal prophylaxis. The ESC update begins to address this for patients with prosthetic valves undergoing cardiac or vascular interventions with recommendations for MRSA carrier nasal screening and pre-procedural mupirocin and chlorhexidine. Interestingly, studies investigating the administration of mupirocin in patients receiving dialysis and in non-cardiothoracic surgical patients have shown a significant reduction in staphylococcal infections, and in particular, staphylococcal bacteremia, especially if patients are carriers of *S. aureus*. Whether administration of peri-procedural mupirocin for invasive procedures would demonstrate a reduction of PVE is an interesting question. The 2023 update further advances anti-staphylococcal measures with the inclusion of *S. aureus* prophylaxis for cardiac device implantation at a level Ia. 

## 8. Cardiac Device-Related Endocarditis

While criteria vary across studies, most experts classify cardiac device-related endocarditis as the presence of a vegetation anywhere along the cardiac valves, endocardial surface, or along the implantable cardiac device (CIED). The infection can arise directly from the subcutaneous device or through bacterial seeding. Globally, there has been increased use of implantable cardiac devices including pacemakers, defibrillators, resynchronization therapy devices, and left ventricular assist devices [21]. This is in part due to expanded indications for implantation as well as an aging population. Of concern, the rates of CDRE continue to increase disproportionately to implantation rates [4]. One study using the Nationwide Inpatient Sample (NIS) data to assess over 4 million pacemaker and defibrillator implantations between 1993 and 2008 found CDRE incidence increased by 210% alongside a 96% increase in device implantation over the same period (Figure 1) [22]. Similarly, another study conducted within the United States showed a year-on-year increase in the proportion of patients developing CDRE relative to the number of implantations each year [23]. CIEDs carry a compounding risk of infection as they require repeated interventions over the lifetime of the patient. Accessing the device pocket for generator changes or other revisions has been found to result in a 3-fold increase in infection risk with each intervention [24,25]. The Prevention of Arrhythmia Device Infection Trial (PADIT) investigated the effect of expanded antimicrobial prophylaxis on 1-year risk of infection and showed no significant reduction compared to standard preoperative antimicrobial prophylaxis during CIED implantation [31,41]. One proposed inference is that a greater portion of CDRE infections develop later, post implantation, as opposed to peri-operatively as previously conceived.

CDRE are now considered to make up approximately 5–10% of all infectious endocarditis [22,42,43]. Notably, CDRE infections present unique and difficult challenges in diagnosis and management. CDRE are often serum-culture negative and may require cultures to be drawn directly from the pocket or extracted leads. Diagnosis of CDRE via echocardiography can be substantially more difficult. Interference from the device’s metal often yields non-diagnostic or inconclusive studies. Moreover, many portions of the devices are not visible with echocardiography. On the other hand, incidental masses including thrombi or non-pathogenic fibrin sheaths are not uncommon and can result in false positive reads and unnecessary interventions [44]. One prospective study conducted over 2 years found transvenous lead masses were present in 14% of patients, with 72% of masses proving to not be infectious [43]. Subsequent imaging modalities are often required in the evaluation of CDRE. In cases of CDRE, more advanced imaging studies such as FDG PET/CT or MSCT are often warranted. 

Clinical management is challenging as the risk of removing a lifesaving device is weighed against the risk of persistent or recurrent infection, each with its own potential morbidity and mortality. A NIS analysis of 59,082 patients who underwent lead extraction due to CIED-related infection found that 10.4% of patients had at least one peri-procedural complication and mortality occurred in 4% of patients [45]. Current 2017 HRS guidelines recommend complete device removal and lead replacement for confirmed cases of CDRE. Multiple studies substantiate HRS’s position due to repeated observations of significantly increased 30-day and one-year mortality with antimicrobial therapy alone. However, guidance on the timing of extraction is lacking and many physicians delay extraction in favor of antibiotic therapy. Few studies exist investigating early versus delayed extraction. Though not standardized, early extraction has been experimentally defined as within 7 days. One retrospective, single-center study consisting of 233 patients who underwent CIED lead extraction found that delayed extraction is associated with increased morbidity and mortality. Of note, the study population has significant differences in baseline comorbidities that may have led to worse outcomes for the delayed cohort, including increased prevalence of heart failure, COPD, diabetes, hypertension, and anemia requiring transfusions. Another investigation using the Nationwide Readmission Database similarly found that lead extraction greater than 7 days was associated with increased in-hospital mortality, and Viganego et al. found that performing lead extraction within 3 days of diagnosis was associated with lower in-hospital mortality [46,47]. Importantly, the nature of these studies may lend to selection bias, with the delayed cohort being significantly sicker at baseline and potentially precluded from early intervention. While guidelines outline reimplantation timing, there is an ongoing need to define and address explantation timing.

## 9. Management

Prior to the advent of antimicrobial management, infective endocarditis was universally fatal [48]. Ethical concerns make randomized control trials less than ideal to tailor effective regimens, with current recommendations significantly shaped by cohort studies and clinical practice. Empiric antibiotics should include coverage against methicillin-resistant *Staphylococcus aureus*, *Streptococcus*, and aerobic Gram-negative bacilli [10]. Subacute presentations warrant additional empiric coverage for HACEK, VGS, and *Enterococci* [10]. Of note, CoNS are typically resistant to methicillin and are seen at an increased rate in patients with healthcare-associated staphylococcal IE [10]. Once an organism is identified, antibiotic coverage can be narrowed. Native valves can be treated for as little as 4 weeks from the first set of negative blood cultures in certain conditions, while prosthetic valves require at least 6 weeks of treatment (Table 1). To improve efficacy, combination intravenous therapy is preferred over.

Current guidelines diverge on the number of minimally recommended inpatient IV antibiotics days—7 days per ESC and 2 weeks per AHA [10,31]. The AHA guidelines argue that outpatient parenteral antibiotic therapy (OPAT) may be pursued only after 2 weeks of inpatient therapy when the greatest risk for septic emboli has subsided and if the following stipulations are met: the patient is low risk for complications such as heart failure and systemic emboli and is free of cardiac conduction abnormalities, valve ring abscess, persistent fever, or persistently positive blood cultures [10]. The ESC endorses similar stipulations including exclusion of patients with cirrhosis, highly difficult to treat organisms, and extra-valvular cardiac complications [31]. Sole use of oral therapy for IE has largely been discouraged and is considered a departure from standard of care over concerns of absorption reliability. However, a randomized, non-inferiority, multi-center trial in Denmark looked at transitioning patients to oral antibiotic treatment after at least 10 days of IV therapy in comparison to OPAT. Parenteral antibiotic regimens were administered per European Society of Cardiology (ESC) guidelines while oral regimens were based on minimal inhibitory concentrations (MIC) for each bacterial species as reported by the European Committee on Antimicrobial Susceptibility Testing (EUCAST). Three of the four components of the primary outcome occurred equally (unplanned cardiac surgery, embolic event, relapse of positive blood cultures) within the two treatment arms. All-cause mortality, the final component, occurred more frequently in the IV treatment group (6.5% vs. 3.5%) (hazard ratio, 0.53; 95% CI, 0.21–1.32). Ultimately, stable patients with left-sided IE caused by *S. aureus*, *Streptococcus*, *E. faecalis*, or CoNS who were transitioned to orally administered antibiotics were found to have non-inferior treatment in comparison to those who continued IV management [49]. Follow-up data from the POET trial 5 years later were similar, with a lower incidence of all-cause mortality in the oral step-down treatment arm versus continued IV treatment (23.4% vs. 35.2%) (hazard ratio, 0.61; 95% CI, 0.42 to 0.88) [50]. Additionally, there was no indication of long-term treatment failure in the oral step-down therapy group. As such, the 2023 ESC guidelines now recommend either parenteral or oral therapy equivocally following completion of initial inpatient IV therapy for qualified patients (Figure 2) [31]. 

Despite adequate antimicrobial treatment, some patients will experience complications related to IE including heart failure, septic emboli, and persistent infection with possible multi-organ dysfunction. Surgical management is required in nearly half of patients with infective endocarditis due to the presence of or increased risk for severe complications [35]. Early surgical management transpires while the patient is still receiving antibiotic treatment and is reserved for patients with severe valvular dysfunction resulting in acute heart failure, those with destructive penetrating lesions, heart block, or aortic/annular abscesses, as well as for cases of persistent infection (fever lasting > 5–7 days or inability to clear cultures) despite appropriate antimicrobial therapy [10]. Other considerations for early surgical management include IE caused by highly resistant organisms or fungi, persistent/enlarging vegetations, recurrent emboli, and mobile vegetations ≥ 10 mm [10]. While previous ESC guidelines recommended urgent surgery at 15 mm, the new revisions call for urgent surgery for a vegetation size ≥ 10 mm. Surgery during the active phase of infection is associated with significant risk and is undertaken when needed to avoid rapidly progressive heart failure, irreversible structural damage, and to prevent systemic embolism [51]. Systemic emboli are estimated to occur in up to 50% of cases of IE. The risk is highest before the initiation of antibiotics and through the first 2 weeks of therapy. The incidence of embolism dramatically reduces to 6–21% after initiation of antibiotic therapy [35]. The role of early surgical management in preventing emboli is controversial. The data supporting early surgical management is limited, with few randomized trials conducted to examine the role of early valvular surgery in the management of IE. One study, with 76 patients who had severe valve regurgitation, left-sided NVE, and vegetations >10 mm stratified to early surgery within 48 h or conventional treatment, demonstrated a reduction of in-hospital deaths and embolic events (3% vs. 23%) [52]. Notably, participants were young with limited comorbidities and had a lower operative risk compared to the typical IE patient. 

In patients with definite CDRE infection, complete device and lead removal are recommended without delay. Such patients require antimicrobial therapy for 10–14 days following device removal for pocket site infection and at least 14 days after removal for bacteremia, with possible extension up to 6 weeks in the presence of septic emboli or PVE [31,53]. Complicated infections (i.e., septic thrombophlebitis, endocarditis, osteomyelitis, or persistent bacteremia) require additional antimicrobial therapy of at least 4–6 weeks in duration. Other indications for complete device and lead removal are CIED pocket infection (skin adherence, abscess formation, device erosion, chronic draining sinus without evident involvement of the transvenous portion of the lead system), valvular endocarditis even without certain involvement of the device and/or leads, occult staphylococcal bacteremia, and persistent occult Gram-negative bacteria. The 2023 ESC guidelines also recommend, at a level IIa, complete device and lead removal for any persistent bacteremia despite completion of appropriate antimicrobial therapy [31]. Assessment for continued CIED indication should be carried out for all patients potentially requiring device removal. In patients where a new device is necessary, attempting to place the device on the contralateral side is reasonable; otherwise, a transvenous lead can be tunneled to a subcutaneously placed device in the abdomen [53]. Reimplantation should be delayed until clinical signs of infection have resolved and blood cultures are negative for at least 72 h in the absence of visible vegetations or 2 weeks for visualized vegetations [31]. 

Long-term follow-up is aimed at preventing recurrent infection and evaluating for delayed sequela. One study reported an annual risk of recurrent IE at 0.62% per patient per year [54]. Those with prosthetic valve endocarditis, liver cirrhosis, parenteral drug use, and those who did not undergo surgical management had an increased risk of a second IE episode [54].

## 10. Conclusions

Infective endocarditis remains a significant challenge. The changing profile of the disease, from pathogenicity to population at risk, poses significant complexities in prevention, diagnosis, and management. The growing presence of prosthetic valves and implantable cardiovascular electronic devices necessitates a high index of suspicion for disease, new modes of evaluation, and evolving guidance for management. Technological advancements in imaging are readily being made and have demonstrated their utility in early and more accurate diagnosis, especially in cases of PVE and CDRE. As they become more widely available and utilized, it is likely that the future of IE imaging will be multimodal. Diagnostic criteria should be modified to reflect findings from new imaging modalities to prompt a more complete diagnosis.

## 11. Future Directions

Given the rising proportion of PVE and CDRE, additional studies are warranted to investigate the application of use-specific multimodal imaging. These studies should guide the development of clearly defined indications for additional imaging studies. Moreover, investigations into whether these new modalities should be undertaken earlier in the clinical course, and not just as sequential imaging, to potentially provide earlier detection are needed. Despite evidence supporting use of alternative imaging, their adoption in the clinical diagnosis and management of IE has been limited. Further investigation into the limiting factors is warranted especially in light of recent guidelines on diagnostic protocols. As new cases are identified, there is a need to further develop risk stratification tools for early versus late surgical management. With the rising incidence of PVE, studies assessing primary PVE risk and IE recurrence by valve type are certainly needed. Additionally, for those with prosthetic valves or CIEDs, identifying additional anti-staphylococcal measures is key. Lastly, further guidance for the timing of CIED extraction in cases of CDRE is needed. 

## Figures and Tables

**Figure 1 diagnostics-13-03086-f001:**
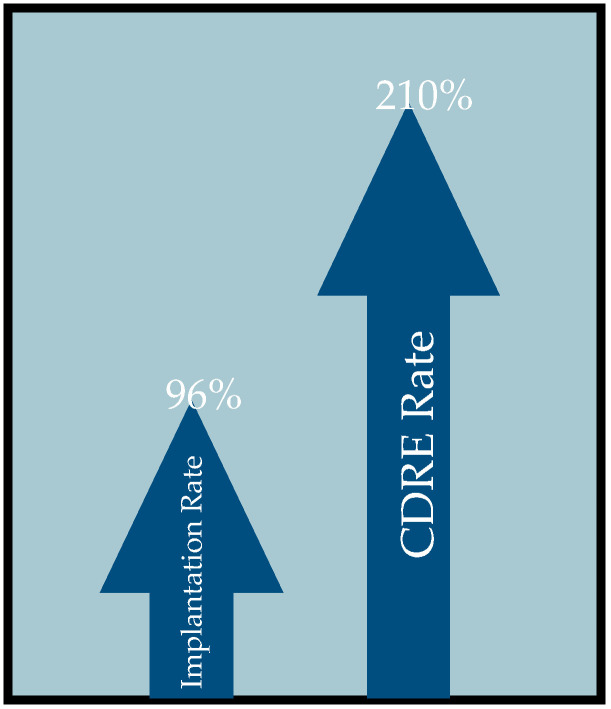
Comparative CIED implantation vs. infection rate [22].

**Figure 2 diagnostics-13-03086-f002:**
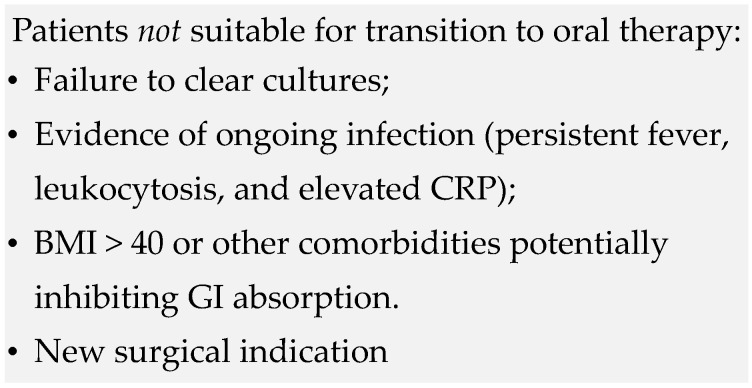
Parenteral step-down eligibility [31].

**Table 1 diagnostics-13-03086-t001:** Recommended antibiotic therapies [10].

Pathogen	Recommended Therapy
Native Valve	
VGS and *Streptococcus gallolyticus* (*bovis*) PCN susceptible	Penicillin G 12–18 million U/24 h IV either continuously or in 4 or 6 equally divided doses for 4 weeks OR Ceftriaxone 2 g/24 h IV/IM in 1 dose for 4 weeks. Alternatively Penicillin G 12–18 million U/24 h IV either continuously or in 6 equally divided doses for 2 weeks or Ceftriaxone 2 g/24 h IV/IM in 1 dose for 2 weeks PLUS Gentamicin 3 mg/kg/24 h IV or IM in 1 dose for 2 weeks and vancomycin 30 mg/kg/24 h IV in 2 equally divided doses)
VGS and *Streptococcus gallolyticus* (*bovis*) PCN resistant	Penicillin G 24 million u/24 h IV either continuously or in 4–6 equally divided doses for 4 weeks PLUS Gentamicin 3 mg/kg per 24 h IV or IM in 1 dose for 2 weeks AND vancomycin 30 mg/kg/24 h IV in 2 equally divided doses for 4 weeks
*Staphylococcus* methicillin-susceptible	Nafcillin or oxacillin 12 g/24 h in 4–6 equally divided doses for 6 weeks (complicated right-sided and left-sided IE; 2 weeks for uncomplicated right-sided) OR cefazolin 6 g/24 h IV in 3 equally divided doses (if anaphylactoid hypersensitivity to β-lactams use vancomycin).
*Staphylococcus* methicillin-resistant	Vancomycin 30 mg/kg/24 h in 2 equally divided doses for 6 weeks OR daptomycin ≥8 mg/kg/dose for 6 weeks.
*Enterococcus* susceptible to penicillin and gentamicin	Ampicillin 2 g IV every 4 h for 4 weeks (sx < 3 months) or 6 weeks (sx >3 months) OR Penicillin G 18–30 million U/24 h IV either continuously or in 6 equally divided doses for 4 weeks PLUS Gentamicin 3 mg/kg ideal body weight in 2–3 equally divided doses. An alternative regimen is double β-lactam Ampicillin 2 g IV every 4 h for 6 weeks PLUS Ceftriaxone 2 g every 12 h for 6 weeks (recommended for patients with creatinine clearance < 50 mL/min).
*Enterococcus* susceptible to penicillin and resistant to Aminoglycosides or Streptomycin-Susceptible Gentamicin-Resistant	Double β-lactam Ampicillin 2 g IV every 4 h PLUS Ceftriaxone 2 g IV every 12 h for 6 weeks. Alternative for Streptomycin-Susceptible Gentamicin-Resistant includes Ampicillin 2 g every 4 h for 4 weeks OR Penicillin G 18–30 million U/24 h IV either continuously or in 6 equally divided doses PLUS Streptomycin 15 mg/kg ideal body weight/24 h IV/IM in 2 equally divided doses for 4 weeks (Patients with creatinine clearance < 50 mL/min or develop creatinine clearance < 50 mL/min during treatment should be treated with double–β-lactam regimen. Patients with abnormal cranial nerve VIII function should be treated with double–β-lactam regimen).
*Enterococcus* Vancomycin- and Aminoglycoside-susceptible Penicillin-Resistant unable to tolerate β-lactam:	Vancomycin 30 mg/kg/24 h IV in 2 equally divided doses PLUS Gentamicin 3 mg/kg/24 h IV/IM in 3 equally divided doses for 6 weeks.
*Enterococcus* Penicillin-, aminoglycoside-, and vancomycin-resistant:	Linezolid 600 mg IV/PO every 12 h for >6 weeks OR Daptomycin 10–12 mg/kg per dose for >6 weeks
HACEK	Ceftriaxone 2 g/24 h IV/IM in 1 dose for 4 weeks OR Ampicillin 2 g IV every 4 h for 4 weeks OR Ciprofloxacin 1 g/24 h PO or 800 mg/24 h IV in 2 equally divided doses for 4 weeks.
Prosthetic valve	
VGS and *Streptococcus gallolyticus* (*bovis*) PCN susceptible:	Penicillin G 24 million U/24 h IV either continuously or in 4–6 equally divided doses for 6 weeks OR Ceftriaxone 2 g/24 h IV or IM in 1 dose for 6 weeks PLUS Gentamicin 3 mg/kg/24 h IV or IM in 1 dose for 2 weeks (Vancomycin 30 mg/kg per 24 h IV in 2 equally divided doses for patients intolerant of PCN or CTX).
*Staphylococcus* methicillin-susceptible	Nafcillin or oxacillin 12 g/24 h in 4–6 equally divided doses for ≥6 weeks (Vancomycin should be used for immediate-type hypersensitivity reactions to β-lactam antibiotics) PLUS cefazolin 6 g/24 h IV in 3 equally divided doses for ≥6 weeks (cefazolin may be substituted for non-immediate-type hypersensitivity reactions to penicillins) PLUS Gentamicin 3 mg/kg/24 h IV or IM in 2 or 3 equally divided doses for 2 weeks.
*Staphylococcus* methicillin-resistant	Vancomycin 30 mg/kg/24 h in 2 equally divided doses for ≥6 weeks PLUS Rifampin 900 mg/24 h IV/PO in 3 equally divided doses for ≥6 weeks PLUS Gentamicin 3 mg/kg/24 h IV/IM in 2 or 3 equally divided doses for 2 weeks.
*Enterococcus* susceptible to penicillin and gentamicin	Ampicillin 2 g IV every 4 h for 6 weeks (sx < 3 months) or 6 weeks (sx > 3 months) OR Penicillin G 18–30 million U/24 h IV either continuously or in 6 equally divided doses for 6 weeks PLUS Gentamicin 3 mg/kg ideal body weight in 2–3 equally divided doses. An alternative regimen is double β-lactam Ampicillin 2 g IV every 4 h for 6 weeks PLUS Ceftriaxone 2 g every 12 h for 6 weeks (recommended for patients with creatinine clearance < 50 mL/min).
*Enterococcus* susceptible to penicillin and resistant to Aminoglycosides or Streptomycin-Susceptible Gentamicin-Resistant:	Double β-lactam Ampicillin 2 g IV every 4 h PLUS Ceftriaxone 2 g IV every 12 h for 6 weeks. Alternative for Streptomycin-Susceptible Gentamicin-Resistant includes Ampicillin 2 g every 4 h for 6 weeks OR Penicillin G 18–30 million U/24 h IV either continuously or in 6 equally divided doses PLUS Streptomycin 15 mg/kg ideal body weight/24 h IV/IM in 2 equally divided doses for 6 weeks (Patients with creatinine clearance < 50 mL/min or develop creatinine clearance < 50 mL/min during treatment should be treated with double–β-lactam regimen. Patients with abnormal cranial nerve VIII function should be treated with double–β-lactam regimen).
*Enterococcus* Vancomycin- and Aminoglycoside-susceptible Penicillin-Resistant unable to tolerate β-lactam	Vancomycin 30 mg/kg/24 h IV in 2 equally divided doses PLUS Gentamicin 3 mg/kg/24 h IV/IM in 3 equally divided doses for 6 weeks.
*Enterococcus* Penicillin-, aminoglycoside-, and vancomycin-resistant	Linezolid 600 mg IV/PO every 12 h for >6 weeks OR Daptomycin 10–12 mg/kg per dose for >6 weeks (Linezolid use may be associated with potentially severe bone marrow suppression. Patients should be treated by a care team including specialists in infectious diseases, cardiology, cardiac surgery, and clinical pharmacy. Of note, cardiac valve replacement may be necessary for cure).
HACEK	Ceftriaxone 2 g/24 h IV/IM in 1 dose for 6 weeks OR Ampicillin 2 g IV every 4 h for 6 weeks OR Ciprofloxacin 1 g/24 h PO or 800 mg/24 h IV in 2 equally divided doses for 6 weeks.

## Data Availability

Not applicable.

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
