# Peer review of "Contemporary Features and Management of Endocarditis"

_diagnostics, 2023, doi:10.3390/diagnostics13193086_

Round 1
Reviewer 1 Report
Dear authors
This manuscript has evaluated Contemporary features and management of endocarditis
The title of the manuscript is suitable
Abstract
The writing of sentences is appropriate. Please add one or two sentences about main findings of the study.
3 -the introduction is very small, please expended
Please see this papers
· Hubers SA, DeSimone DC, Gersh BJ, Anavekar NS. Infective Endocarditis: A Contemporary Review. Mayo Clin Proc. 2020 May;95(5):982-997. doi: 10.1016/j.mayocp.2019.12.008. Epub 2020 Apr 13. PMID: 32299668.
· Feature | Infective Endocarditis: Words of Caution For Management
· Bayer AS, Bolger AF, Taubert KA, Wilson W, Steckelberg J, Karchmer AW, Levison M, Chambers HF, Dajani AS, Gewitz MH, Newburger JW. Diagnosis and management of infective endocarditis and its complications. Circulation. 1998 Dec 22;98(25):2936-48.
Epidemiology
According this paper is review ,please see all paper in this area
Manuscript body sections
Addition of one or more figures and text findings can make it more understandable in a shorter time for readers instead of studying a whole text in a review paper.
9. Management: is very useful
Conclusion
It is suitable,
References
There is lacking of studies from 2022 and 2023 in the references. If needed, these studies can be added to make the literature more powerful.
With best regards
Extensive editing of English language required
Author Response
Thank you for your review. See attached.

Reviewer 2 Report
Summary
This narrative review evaluates various aspects of infective endocarditis, including changes over time of populations and valves/devices at risk, emerging diagnostic approaches and therapeutic challenges.
General Comment
This is a well-written and up-to-date narrative review. Some modifications, as shown below, are needed.
Specific Comments
The affiliations should be detailed.
Abstract could be improved with some detailed information.
Introduction: The topics that you mean to address in this review to provide an updated picture from a diagnostic and therapeutic point of view should be better reported.
Page 2: Ref 13 is followed by Refs 27-38, and then again by Ref 14. Please, check.
In the text, some acronyms should be defined first.
There are some repetitions in the text.
There are some typos.
Please, check each reference.
Minor editing of English language required
Author Response
Thank you. See attached.

Reviewer 3 Report
The manuscript is a review of a very wide topic of endocarditis. Some issues are only mentioned, others like CDRE described in detail. Maybe it would be beneficial to focus on one of endocarditis problems, instead of trying to deal with all, which is very difficult to perform in such a form of paper. New ESC guidelines on management of endocarditis will occur during next week or two, during ESC, so it is difficult to assess how the paper adheres to newest recommendations.
Author Response
Thank you kindly for your review and suggestions. We have reviewed the 2023 ESC guidelines that were made available ahead of print and incorporated the changes. We were not able to obtain a complete copy of the new Duke Criteria, but from review of the available abstract we have updated the text with the highlighted changes including removal of the 30 minute culture draw time.
Round 2
Reviewer 1 Report
It is suitable,
Author Response
Thank you for your time and re-revew
Reviewer 3 Report
Closer adherence to the newest ESC guidelines should be performed - point by point. Particularly, changes in preventions and pacemaker reimplantation should be implemented.
minor spelling corrections
Author Response
Additional evaluations of the 2023 ESC guidelines were included for PVE, CDRE, surgical management, and antibiotic prophylaxis.